# High Diversity and Prevalence of Rickettsial Agents in *Rhipicephalus microplus* Ticks from Livestock in Karst Landscapes of Southwest China

**DOI:** 10.3390/microorganisms13040765

**Published:** 2025-03-27

**Authors:** Ya-Ting Liu, Yi-Fei Wang, Ming-Zhu Zhang, Dai-Yun Zhu, Yi Sun, Cai-Wei Gong, Lin Zhan, Xiao-Ming Cui, Wu-Chun Cao

**Affiliations:** 1State Key Laboratory of Pathogen and Biosecurity, Academy of Military Medical Sciences, Beijing 100071, China; 15030364511@163.com (Y.-T.L.);; 2Institute of EcoHealth, School of Public Health, Cheeloo College of Medicine, Shandong University, Jinan 250012, China; wangyf419@163.com (Y.-F.W.);; 3Animal Husbandry Development Center of Qiannan Buyei and Miao Autonomous Prefecture, Duyun 558000, China; 4National Health Commission Key Laboratory of Pulmonary Immunological Diseases, Guizhou Provincial People’s Hospital, Guiyang 550001, China

**Keywords:** karst landscapes, tick-borne microorganisms, spotted fever group *Rickettsia*, *Anaplasma*, *Ehrlichia*, *Rhipicephalus microplus*

## Abstract

Ticks and tick-borne pathogens pose a significant threat to human and animal health, yet the diversity and prevalence of tick-borne microorganisms in karst regions remains inadequately explored. In October 2023, a total of 274 *Rhipicephalus microplus* ticks were collected from livestock in Guizhou Province, which boasts the largest karst area in China. Pathogen identification was subsequently performed using PCR amplification, Sanger sequencing, and phylogenetic analysis. High microbial diversity was noted, with five bacterial species from the order Rickettsiales detected, including those from the genera *Rickettsia* (family *Rickettsiaceae*), *Anaplasma*, and *Ehrlichia* (family *Anaplasmataceae*). The overall prevalence of infection with at least one pathogen was remarkably high at 94.5%. The highest positive rate was observed for *Candidatus* Rickettsia jingxinensis at 90.9%. A novel *Ehrlichia* species, provisionally designated as *Candidatus* Ehrlichia carsus, was identified with a positive rate of 16.8%. In addition, *Anaplasma marginale*, *Ehrlchia minasensis* and *Ehrlichia canis* were detected in 15.3%, 4.7% and 1.5%, respectively. The co-infections involving two or three rickettsial species were observed in 34.3% ticks. These findings highlight the high diversity and prevalence of tick-borne rickettsial agents in the karst area, underscoring the need for enhanced surveillance and effective tick control to mitigate disease risks to both humans and livestock.

## 1. Introduction

Ticks are obligate blood-feeding arthropods and important vectors of a wide range of pathogens, including viruses, bacteria, and protozoa, which pose significant risks to both human and animal health [1,2,3,4]. Throughout their life cycle, from larva to adult, ticks depend on blood feeding for development, which enables them to acquire and transmit pathogens from hosts at every stage. Furthermore, many pathogens can be transmitted transovarially and transstadially, enabling ticks to function not only as vectors but also as reservoirs for these infectious agents [5]. The dual role further underscores the significance of ticks in pathogen transmission and complicates control efforts. Ticks pose a significant threat to the livestock industry. Their direct impact includes blood-sucking, leading to anemia and weight loss, while their indirect harm lies in transmitting pathogens, compromising animal husbandry, reducing productivity, and ultimately causing substantial economic losses [6].

The order Rickettsiales comprises a group of Gram-negative bacteria that includes three main lineages: *Rickettsiaceae*, *Anaplasmataceae*, and *Candidatus* Midichloriaceae [7,8]. With the continuous discovery of new pathogenic species and their associated diseases, the order Rickettsiales has garnered widespread attention [9]. For instance, *Anaplasma capra*, a zoonotic pathogen initially reported in China in 2015, can infect humans, goats, and sheep [10]. In 2022, *Rickettsia aeschlimannii* was first identified in patients in Xinjiang [11], and in 2024, *Rickettsia solvaca* was detected for the first time in patients in Inner Mongolia [12]. These findings highlight the ongoing emergence and spread of tick-borne rickettsial pathogens, emphasizing the need for continued surveillance and research to mitigate their impact on human and animal health.

Karst landscapes are widely distributed across the globe, found in regions such as the Balkan Peninsula, southwest China, Malaysia, and Indonesia. Karst areas of China account for approximately 15.6% of the global karst area [13]. Among these, Guizhou Province has the largest karst area, covering about 73% of the province’s land. The karst landscape of southwest China has vast primary forests and highly diverse ecosystems, making it recognized as one of the world’s biodiversity hotspots. The region’s unique ecological conditions and geographical features provide an ideal habitat for ticks [14]. However, research on ticks and tick-borne pathogens in karst regions remains limited, lacking a comprehensive understanding of the diversity and infection rates of tick-borne agents in this area.

Guizhou Province, characterized by its typical karst landscapes and well-developed animal husbandry, is a high-risk area for tick-borne diseases. Previous studies have identified various tick-borne pathogens in some areas of Guizhou, including *Rickettsia monacensis*, *Candidatus* Rickettsia jingxinensis (Ca. R. jingxinensis), *Anaplasma capra*, and Jingmen tick virus [15,16,17]. However, the unique ecological environment of the karst landscape may determine the presence of tick-borne agents. Therefore, the present study focuses on the karst regions of Guizhou. The objective is to elucidate the composition and distribution patterns of tick-borne pathogens in this region. Due to local environmental constraints and the limited number of sampling sites, parasitic ticks were collected from livestock in six counties. Despite the limited sample size, we conducted an epidemiological investigation of tick-borne pathogens, aiming to provide scientific evidence and data support for the prevention and control of tick-borne diseases in the region.

## 2. Materials and Methods

### 2.1. Sample Collection

In October 2023, both blood-feeding and host-questing ticks were collected from livestock at nine sampling sites within the karst region of Guizhou Province. The tick species were systematically identified by an entomologist (Yi Sun) through detailed morphological analysis, including to species, developmental stage, and sex.

### 2.2. DNA Extraction

Each tick was processed as an individual sample for DNA extraction. Ticks were placed into 1.5 mL centrifuge tubes with 2–3 steel beads and 200 µL RNase-free water. The tubes were then homogenized in a tissue grinder under the following conditions: 4 °C, 55 Hz for 300 s. Following homogenization, DNA was extracted according to the manufacturer’s protocol for the TaKaRa MiniBEST Viral RNA/DNA Extraction Kit Ver.5.0 (TaKaRa Bio Inc, Kusatsu, Japan). The DNA was then eluted in 200 µL RNase-free water for subsequent analysis.

### 2.3. PCR Assays and Sequencing

Nucleic acid samples extracted from ticks were used as templates for PCR amplification targeting specific gene fragments: the outer membrane protein A (*ompA*) gene, the *17 kDa* gene, and the citrate synthase (*gltA*) gene for *Rickettsia*; the *gltA* gene, the heat shock protein (*groEL*) gene, and the major surface protein 4 (*msp4*) gene for *Anaplasma*; the 16S rRNA gene, *gltA* gene, and *groEL* gene for *Ehrlichia*; the 18S rRNA gene for *Babesia* and *Theileria*; the *16S rRNA* and the *5S-23S rRNA* gene for *Borrelia* (Appendix A). Samples showing positive detection for all three gene fragments of *Rickettsia*, *Anaplasma*, and *Ehrlichia* were considered positive. All PCR-positive amplicons were subsequently sent to Beijing Tianyi Huiyuan Biotechnology Co., Ltd. for paired-end sequencing. The relevant sequences have been uploaded to GenBank (Appendix A).

### 2.4. Phylogenetic Analysis

The obtained sequences were initially compared using the BLAST tool from NCBI (http://www.ncbi.nlm.nih.gov/BLAST, accessed on 15 January 2025). The reference sequences from different strains in GenBank were used for concatenated tree construction (Appendix A). The sample sequences were then aligned with reference sequences using MAFFT (v7.487) software [18], followed by trimming of unreliable regions using Gblocks (v0.91b) [19]. The aligned sequences were subsequently concatenated. Phylogenetic trees based on individual gene fragments and concatenated genes were constructed using IQ-TREE (v2.1.4), with the optimal substitution models selected. Trees were generated using the maximum likelihood (ML) method. To assess the reliability of the phylogenetic tree, ultrafast bootstrap analysis with 1000 iterations was performed [20]. The resulting phylogenetic tree was annotated and visualized using the iTOL online tool (v7.1) (https://itol.embl.de/, accessed on 16 February 2025) [21]. Finally, genetic distances were calculated using Mega 11 [22].

### 2.5. Statistical Analyses

The count data are presented as n (%), with 95% confidence intervals (95% CI) calculated using the Wilson score method, which provides more accurate interval estimates, particularly for small sample sizes or proportions close to 0 or 1. Chi-square tests (χ^2^) were conducted to assess differences in positivity rates between groups [23]. When expected frequencies were low, Fisher’s exact test was considered as an alternative. For comparing positive rates of *Rickettsia*, *Anaplasma*, and *Ehrlichia*, paired sample analyses were performed, with Cochran’s Q test used to evaluate overall differences and McNemar’s test applied for pairwise comparisons. All statistical analyses were conducted using SPSS 22.0 software, applying two-sided tests with a significance level set at *p* < 0.05.

## 3. Results

### 3.1. Tick Sampling

A total of 274 adult tick samples were collected from nine sampling sites in the karst region of Guizhou Province (Figure 1). Morphological identification confirmed that all the ticks were *Rhipicephalus microplus* (*R. microplus*), comprising 200 females and 74 males, with 183 blood-feeding ticks and 91 host-questing ticks. Except for one site, where goats served as tick hosts, the remaining sites hosted cattle.

### 3.2. Phylogenic Analysis of Different Tick-Borne Pathogens

Molecular screening for additional pathogens, including *Babesia*, *Theileria*, and *Borrelia* yielded negative results. Our findings were exclusively focused on members of the order Rickettsiales, which included species from the spotted fever group of the family *Rickettsiaceae*, as well as two genera within the family *Anaplasmataceae*: *Anaplasma* and *Ehrlichia*.

For *Rickettsia*, 249 samples tested positive for the *ompA*, *gltA*, and *17 kDa* gene fragments, exhibiting identical sequences across all positive samples. The *ompA* gene sequence of these samples exhibited 100% identity with *Candidatus* Rickettsia jingxinensis identified in *Haemaphysalis longicornis* from Shaanxi Province, China (GenBank accession number: MH932061.1) [24] and *Haemaphysalis spinigera* from Idukki, Kerala, India (GenBank accession number: MN463682.1) (Appendix A). The *gltA* gene sequence showed 100% identity with Ca. R. jingxinensis detected in *Rhipicephalus microplus* from China (GenBank accession number: MW114883.1) [25] (Appendix A). The 17 *kDa* gene sequence also exhibited 100% identity with *Candidatus* Rickettsia jingxinensis detected in China (GenBank accession number: OR801788.1) (Appendix A). Additionally, in the phylogenetic tree constructed from the concatenated three gene fragments, the detected *Rickettsia* clustered with Ca. R. jingxinensis detected in Shaanxi, China in 2019, and with the strain identified in India in 2020 (Figure 2). These findings collectively suggest that the *Rickettsia* identified in the positive samples is Ca. R. jingxinensis, yielding a positivity rate of 90.9% (95% CI: 86.9–93.7). No significant differences were observed in positivity rates of Ca. R. jingxinensis between genders or blood-feeding statuses.

For *Anaplasma*, a total of 42 samples tested positive for the *gltA*, *groEL*, and *msp4* gene fragments. The *gltA* gene sequences from the 42 positive samples were identical and showed 100% identity with *Anaplasma marginale* (*A. marginale*) detected in *R. microplus* from Colombia (GenBank accession number: MT722102.1) [26] (Appendix A). The *groEL* gene sequences from 32 of the positive samples displayed 100% identity and grouped with *A. marginale* detected in *Bos taurus* in Brazil (GenBank accession number: CP023730.1), while the remaining 10 sequences were identical and clustered with *A. marginale* from *R. microplus* in China (GenBank accession number: KX987395.1) [27] (Appendix A). The *msp4* gene sequences exhibited 99.8–100% identity and clustered phylogenetically with *A. marginale* strains from cattle in Malaysia (GenBank accession number: MT173810.1) and Brazil (GenBank accession number: MK570464.1) [28] (Appendix A). Phylogenetic analysis further confirmed that the species identified in this study was *A. marginale* (Figure 3), with an overall positivity rate of 15.3% (95% CI: 11.5–20.1). The positivity rate was significantly higher in male ticks compared to female ticks (36.5% vs. 7.5%, *p* < 0.001) and notably lower in blood-feeding ticks compared to host-questing ticks (7.7% vs. 30.8%, *p* < 0.001).

For *Ehrlichia*, a total of 63 samples tested positive for the 16S rRNA, *gltA*, and *groEL* gene fragments. Further phylogenetic analysis identified three distinct *Ehrlichia* species. In four positive samples, *Ehrlichia canis* (*E. canis*) was identified, with its 16S rRNA, *gltA*, and *groEL* gene sequences clustering with those of *E. canis* detected in *R. microplus* ticks from Hubei province of China (GenBank accession numbers: KX987326.1, MW428300.1, MW428319.1) [27,29], showing 100% sequence similarity, respectively (Appendix A). In 13 positive samples, the gene fragments were identical across all three gene segments, sharing 100% sequence identity with *Ehrlichia minasensis* (*E. minasensis*) detected in Brazilian cattle (GenBank accession numbers: GCA_004181775.1, GCA_000825765.1) [30] and in *R. microplus* ticks from Hainan province of China (GenBank accession numbers: OR835907.1, OR555731.1, OR555740.1) [31] (Appendix A). The remaining 46 positive samples exhibited 99.8–100% homology with each other for the 16S rRNA gene fragment and also shared 99.8–100% similarity with uncultured *Ehrlichia* sp. detected in *R. microplus* ticks from Hainan, China (GenBank accession numbers: OR835912.1, OR835909.1) [31] and *Hyalomma anatolicum* ticks from Pakistan (GenBank accession number: MN726921.1) [32], clustering together in the same branch of the phylogenetic tree (Appendix A). The *gltA* gene sequences from these samples showed 100% homology with uncultured *Ehrlichia* sp. found in *R. microplus* ticks from Wuhan and Anhui provinces of China (GenBank accession numbers: KX987356.1, OQ185256.1) [27,33], grouping in the same clade (Appendix A). Similarly, the *groEL* gene fragment exhibited 100% sequence identity with uncultured *Ehrlichia* sp. detected in *Rhipicephalus annulatus* ticks from Sri Lanka and *R. microplus* ticks from Anhui, China (GenBank accession numbers: MZ970589.1, OQ185235.1) [23,34], clustering in the same branch (Appendix A). Based on these findings, we constructed a concatenated phylogenetic tree using all three gene fragments (Figure 4), which confirmed that, apart from *E. canis* and *E. minasensis*, the remaining 46 positive samples formed a distinct branch, suggesting the presence of a novel *Ehrlichia* species. Since this new member was found in a karst landscape region, we propose the provisional name *Candidatus* Ehrlichia carsus. The overall positivity rate for *Ehrlichia* detection in ticks was 23.0% (95% CI: 18.4–28.3). The positivity rate for *Ehrlichia* was significantly higher in female ticks compared to male ticks (27.5% vs. 10.8%, *p* < 0.004). Additionally, the positivity rate in blood-feeding ticks was significantly higher than that in host-questing ticks (29.0% vs. 11.0%, *p* < 0.001).

### 3.3. Prevalence of Different Tick-Borne Pathogens

Overall, in the 274 tick samples analyzed, the positive rate of *Rickettsia* was 90.9% (95% CI: 86.9–93.7), while *Anaplasma* and *Ehrlichia* were detected at rates of 15.3% (95% CI: 11.5–20.1) and 23.0% (95% CI: 18.4–28.3), respectively. Among these, 259 ticks (94.5%, 95% CI: 91.2–96.7) were found to be infected with at least one pathogen (Table 1). No significant differences in the positivity rates of tick-borne pathogens were observed between the different regions. Furthermore, the co-infection rate of two or more pathogens was 34.3% (95% CI: 28.9–40.1), primarily involving co-infections of *Rickettsia* with *Anaplasma* and *Rickettsia* with *Ehrlichia* (Table 2). These findings highlight the high prevalence of *Rickettsia* in tick populations and the frequent occurrence of co-infections, underscoring the complexity of tick-borne pathogen transmission.

## 4. Discussion

In this study, we screened *R. microplus* for pathogens and identified several members of the order Rickettsiales, including a novel *Ehrlichia* species that has not been previously reported in other karst regions. Notably, no *Borrelia*, *Babesia*, or *Theileria* species were detected. These findings highlight the critical need for ongoing monitoring of *rickettsial* infections in karst landscapes, as well as the importance of continuous surveillance of local arthropods, mammals, and humans to better understand and mitigate the risks posed by emerging tick-borne pathogens.

All tick samples collected in this study were *R. microplus*, whereas research in European karst regions has primarily focused on *Ixodes ricinus*. For example, Blažeková et al. [35] studied tick samples from the Slovak Karst National Park and identified several species, including *Ixodes ricinus*, *Haemaphysalis concinna*, *Haemaphysalis inermis*, *Dermacentor reticulatus*, and *Dermacentor marginatus*. They also detected pathogens such as *Anaplasma* spp., *Bartonella* spp., *Rickettsia* spp., *Babesia* spp., and *Theileria* spp. Similarly, studies in other karst regions of Slovakia have shown that *Ixodes ricinus* was the predominant tick species, harboring a diverse array of tick-borne pathogens, with the highest infection rate for *Borrelia burgdorferi* s.l., reaching 12.43% [36]. Additionally, Susnjar et al. [37] detected *Borrelia* spp. in *Ixodes ricinus* collected from the Coastal-Karst area in Slovenia. These findings differ from our study, particularly in terms of tick species and the pathogens they carry. This highlights the significant role of the ecological environment and host conditions in shaping tick communities and influencing their capacity for pathogen transmission in karst regions.

We observed differences in pathogen infection rates among *Rickettsia*, *Anaplasma*, and *Ehrlichia* in ticks. For *Rickettsia*, no significant variations in infection rates were found between sex and feeding status, suggesting a relatively uniform transmission potential across these groups. In contrast, *Anaplasma* exhibited distinct differences based on sex and feeding status, with non-blood-feeding ticks showing higher infection rates than blood-feeding ticks, and male ticks exhibiting higher infection rates than female ticks. Conversely, *Ehrlichia* displayed the opposite trend, with blood-feeding ticks having significantly higher infection rates than non-blood-feeding ticks, and female ticks showing higher infection rates than males. Previous studies have also found higher positivity rates for *Borrelia* in female ticks compared to males [38], likely due to differences in blood intake and feeding duration. Additionally, research has shown that *Rhipicephalus sanguineus* with excessive blood feeding exhibit a significantly higher *Ehrlichia* infection rate, potentially due to the increased vectorial capacity associated with higher blood intake [39]. Moreover, we detected a high rate of co-infection with multiple pathogens in ticks, offering important insights into the complex interactions among tick-borne pathogens. Co-infections may alter transmission patterns and contribute to divergent clinical manifestations in hosts, underscoring the necessity for enhanced vector surveillance programs. Such co-infection complexity poses diagnostic challenges and complicates therapeutic interventions in clinical settings. Of particular concern is *Rickettsia*, which was found to co-occur with both *Anaplasma* and *Ehrlichia*, due to its potential to cause severe health issues in humans and animals. The potential interactions between these pathogens within tick vectors may alter transmission efficiency, which could subsequently impact disease severity in infected hosts.

Additionally, we observed a high positivity rate of Ca. R. jingxinensis at 90.9%, which closely aligns with the 96.74% reported in the Qianxinan, Liupanshui, and Bijie regions of Guizhou Province [40], while the positivity rate is significantly higher than the 6.19% seen in the Qiandongnan region [14]. This notably elevated infection rate suggests that Ca. R. jingxinensis may exhibit strong adaptability and transmission capacity in Guizhou, potentially due to favorable ecological conditions, high vector abundance, or host availability. Further investigation is needed to elucidate the underlying factors contributing to this pattern. As a member of the spotted fever group *Rickettsia*, Ca. R. jingxinensis has a broad geographic distribution [25,41]. Clinical cases from South Korea have shown that Ca. R. jingxinensis can infect humans, causing symptoms such as fever, erythema, and eschar, highlighting a potential risk to human health [42]. Therefore, it is crucial to strengthen epidemiological surveillance of both human and animal populations in this region.

*R. microplus* is a primary vector for the transmission of *A. marginale* [43]. In this study, we detected *A. marginale* with a positivity rate of 15.3%, which exhibited high genetic similarity to strains found in Colombia, Brazil, China, and Malaysia. Previous studies have also reported the presence of *A. marginale* across various regions in China, including Hunan, Guangdong (central-southern), Chongqing, Sichuan (southwestern), and Liaoning (northeastern), further confirming the widespread distribution of this pathogen in China [44]. *A. marginale* is the most virulent species of *Anaplasma* affecting cattle, with infected animals commonly showing clinical signs such as fever, lethargy, dark urine, and jaundice. While infections with *A. marginale* can be fatal in adult cattle, many animals remain asymptomatic despite harboring the pathogen [45]. Given the high positivity rate of this pathogen in the region, its significant infectivity and pathogenicity, along with the critical role of livestock farming in the local economy, future efforts should prioritize strengthening surveillance and control measures for *A. marginale* and other related pathogens to reduce their negative impact on livestock production and economic stability.

We also detected three *Ehrlichia* species: a novel strain provisionally named Ca. *E. carsus*, *E. minasensis*, and *E. canis*. The novel Ca. *E. carsus* detected in this study formed a distinct evolutionary branch in the combined phylogenetic analysis, suggesting it may represent an uncharacterized *Ehrlichia* species. Its *16S* gene fragment showed 100% homology with an uncultured *Ehrlichia* species detected on Hainan Island. While the pathogenicity and host range of this species remain undetermined, this discovery adds new evidence of the diversity of pathogens carried by ticks in the karst regions of Guizhou. The positivity rate for *E. minasensis* was 4.7%, which is higher than the 1.68% reported in tick samples from Yingshan County, Hubei [46]. Phylogenetic analysis revealed that the *16S* gene fragment of *E. minasensis* closely resembles that of *E. minasensis* detected on Hainan Island, China. However, the *groEL* and *gltA* gene fragments, along with the combined phylogenetic analysis, showed a strong phylogenetic relationship with *E. minasensis* from Mato Grosso, Brazil [47,48]. *E. canis* primarily infects canines but can also affect humans, typically causing mild or asymptomatic clinical manifestations [49]. Therefore, future research should be performed using whole-genome sequencing and functional experiments to further elucidate the taxonomic classification and biological characteristics of this strain. Notably, both Hainan Island, China, and Mato Grosso, Brazil, are situated in typical karst landscapes [50,51], and this unique ecological environment may provide an ideal habitat for ticks and their pathogens, potentially influencing pathogen transmission. Consequently, future studies should investigate the role of karst ecosystems in shaping tick communities and pathogen transmission, providing valuable insights for tick-borne disease prevention and control.

As an exploratory epidemiological study, this research was conducted at specific sites across six counties in Guizhou Province over a relatively short sampling period. Consequently, the findings may not fully reflect the overall prevalence of tick-borne agents in the karst region of Guizhou. Nevertheless, given the high risk of these diseases in the area, this study provides valuable baseline data for disease prevention and control efforts while laying a crucial foundation for future large-scale research.

## 5. Conclusions

In summary, the prevalence of tick-borne pathogens and the co-infection rate in *R. microplus* ticks from the karst region of Guizhou were relatively high. Therefore, it is crucial to raise awareness among local farmers about effective tick management and disease prevention strategies. Additionally, strengthening monitoring and control efforts targeting ticks and their associated pathogens will be essential not only for protecting livestock health but also for reducing the risk of human infections and safeguarding public health.

## Figures and Tables

**Figure 1 microorganisms-13-00765-f001:**
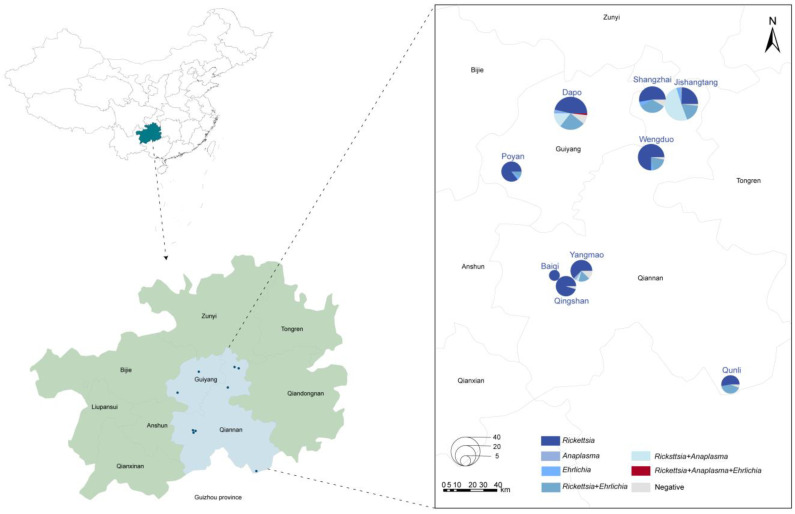
Distribution of tick samples in Guizhou Province, China. Circle size corresponds to the number of ticks collected, while colors represent different pathogens detected in this study.

**Figure 2 microorganisms-13-00765-f002:**
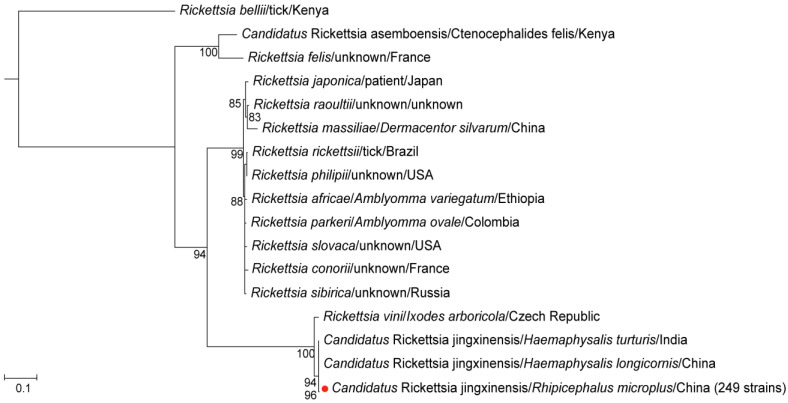
The phylogenetic tree of *Rickettsia* was constructed based on concatenated *ompA*, *gltA*, and *17 kDa* nucleotide sequences using the maximum likelihood (ML) method with 1000 bootstrap replicates. The red dots indicate the evolutionary positions of samples that tested positive for all three gene fragments.

**Figure 3 microorganisms-13-00765-f003:**
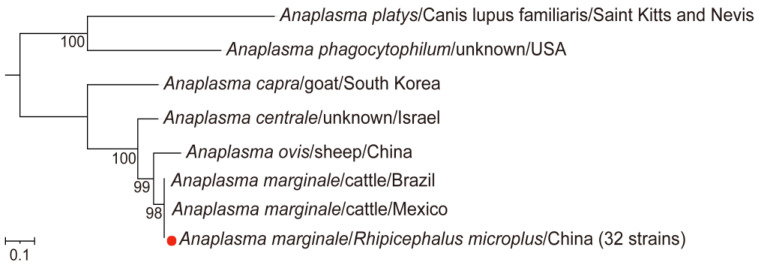
The phylogenetic tree of *Anaplasma* was constructed based on concatenated *gltA, groEL, and msp4* nucleotide sequences using the maximum likelihood (ML) method with 1000 bootstrap replicates. The red dots indicate the evolutionary positions of samples that tested positive for all three gene fragments.

**Figure 4 microorganisms-13-00765-f004:**
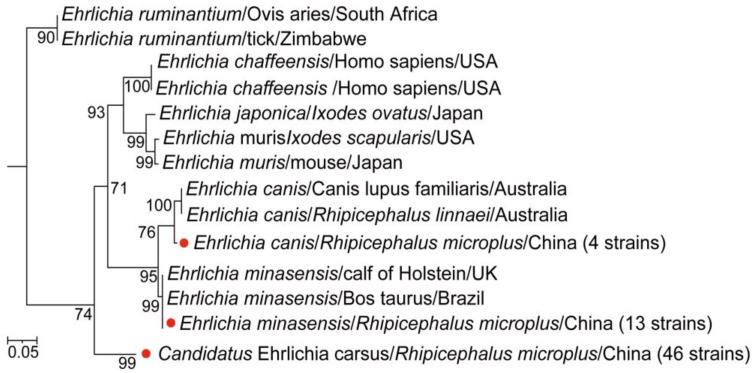
The phylogenetic tree of *Ehrlichia* was constructed based on concatenated *16S rRNA, gltA, and groEL* nucleotide sequences using the maximum likelihood (ML) method with 1000 bootstrap replicates. The red dots indicate the evolutionary positions of samples that tested positive for all three gene fragments.

**Table 1 microorganisms-13-00765-t001:** The prevalence of tick-borne pathogens in ticks from karst area in Guizhou Province of southwest China.

Sample Sites	No. of Tested	Pathogens (%, 95% CI)				
		*Candidatus* Rickettsia Jingxinensis	*Anaplasma marginale*	*Candidatus* Ehrlichia Carsus	*Ehrlichia minasensis*	*Ehrlichia canis*
Wengduo, Guiding	36	34 (94.4, 81.9–98.5)	0	7 (19.4, 9.8–35.0)	1 (2.8, 0.5–14.2)	0
Jishangtang, Weng’an	36	31 (86.1, 71.3–93.9)	0	13 (36.1, 22.5–52.4)	0	0
Shangzhai, Weng’an	57	53 (93.0, 83.3–97.2)	30 (52.6, 39.9–65.0)	4 (7.0, 2.8–16.7)	8 (14.0, 7.3–25.3)	0
Baiqi, Huishui	6	6 (100, 61.0–100)	0	0	0	0
Qingshan, Huishui	21	20 (95.2, 77.3–99.2)	0	0	0	0
Yangmao, Huishui	24	20 (83.3, 64.2–93.3)	2 (8.3, 2.3–25.9)	2 (8.3, 2.3–25.9)	2 (8.3, 2.3–25.9)	0
Qunli, Libo	17	16 (94.1, 73.0–99.0)	0	3 (17.7, 6.2–41.0)	0	4 (23.5, 9.6–47.3)
Dapo, Xifeng	56	49 (87.5, 76.4–93.8)	10 (17.9, 10.0–29.8)	15 (26.8, 17.0–39.6)	1 (1.8, 0.3–9.5)	0
Poyan, Qingzheng	21	20 (95.2, 77.3–99.2)	0	2(9.5, 2.7–28.9)	1 (4.8, 0.9–22.7)	0
Total	274	249 (90.9, 86.9–93.7)	42 (15.3, 11.5–20.1)	46 (16.8, 12.8–21.7)	13 (4.7, 2.5–7.5)	4 (1.5, 0.6–3.7)

**Table 2 microorganisms-13-00765-t002:** Co-infection of different tick-borne pathogens in Guizhou Province.

Pathogen Co-Infections	No. of Positive	Positive Rate (95% CI)
Ca. R. jingxinensis + *A. marginale*	38	13.9% (10.3–18.5)
Ca. R. jingxinensis + *E. canis*	4	1.5% (0.6–3.7)
Ca. R. jingxinensis + *E. minasensis*	12	4.4% (2.5–7.5)
Ca. R. jingxinensis + Ca. *E. carsus*	39	14.2% (10.6–18.9)
Ca. R. jingxinensis + *A. marginale* + Ca. *E. carsus*	1	0.4% (0.1–2.0)
Total	94	34.3% (28.9–40.1)

## Data Availability

The datasets supporting the conclusions of this article are included within the article and Appendix A.

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
