# Peer review of "High Diversity and Prevalence of Rickettsial Agents in Rhipicephalus microplus Ticks from Livestock in Karst Landscapes of Southwest China"

_microorganisms, 2025, doi:10.3390/microorganisms13040765_

Round 1
Reviewer 1 Report
Comments and Suggestions for Authors
It was not clear in the title, abstract or introduction that the authors were limiting their investigation to ticks on livestock or that their survey was only during a single month in 2023. Livestock should be incorporated into all three areas.
The very limited sampling should be noted in the abstract and justified within the paper.
Authorship should only include those Indvidual's who made substantial contributions to the manuscript itself. Other study participants and supporters should be acknowledged. This small study has 17 authors. From what I can tell only 3-4 actually qualify for authorship. These other people should be acknowledged.
The manuscript is not hypothesis driven. Please add a well-supported hypothesis to the introduction
Table 1 is unclear and has some formatting errors. Please revise and reformat. This table also needs a more comprehensive legend/description to explain the data.
There is a discrepancy in the number of samples, line 127 lists the sample number as 259 and other sections indicate 274. This needs to be explained.
A detailed explanation of the many study limitations and their significance should be added to the discussion.
Numerous small grammar and syntax errors exist. I recommend the authors use an English editing service prior to submitting any revisions (for example see lines 60-64).
The references are not consistently formatted to meet journal standards. Most notably the use of capital letters and formatting of journal and article titles.
Comments on the Quality of English Language
Numerous grammar and syntax errors exist. I recommend the authors use an English editing service prior to submitting any revisions (for example see lines 60-64).
"...15.6% of the global."
"...making it recognition..."
Reviewer 2 Report
Comments and Suggestions for Authors
A manuscript (microorganisms-3523889) entitled “High diversity and prevalence of rickettsial agents in Rhipicephalus microplus ticks from karst landscapes of southwest China” authored by Ya-Ting Liu investigates the prevalence of rickettsial agents in R. microplus ticks collected in karst landscapes of southwest China. The items described in the manuscript meet the scope of the journal Microorganisms, in “Microbial ecology”. 48 references are used to construct the article, of which the oldest is from 2007, but the vast majority were published in important scientific journals and in the last 7 years. There are 4 authors with 2 works cited in the article, but without conflict of interest, as they only demonstrate the continuity of their research. The introduction is objective and informative, but at the end, the objective of the work was not clear. This was implied. Even so, I do not suggest its modification, as it innovates the writing style. The methodology is well structured, although straightforward and concise. The statistical tests are robust. In the figures, aiming at the understanding of those who are not familiar with the subject, it is recommended to explain the meaning of the numbers in the keys and what the red ball indicates. Captions should be self-explanatory. In the body of the text, 259 positive samples are mentioned, while in the figure the number described is 249. If the information is different, this should be made clear. For Table 1, better formatting of the rows and columns is suggested. I believe that the font size could be reduced so that the values ​​fit in the rows. The results are simple and straightforward. The discussion is well-prepared, as is the conclusion. To the authors:
In line 23, correct “China, this” to “China, and this”.
In line 73, the entomologist who classified the tick species is mentioned. It is suggested that his professional origin be included in the same field, attesting to his skill in the area. This makes the material more technically sound.
In lines 226 and 233, there are citations with names, without numbering, which is not in line with the journal's rules.
In lines 255 and 256, information is reported about the transmission capacity and adaptation of the etiological agent in the region, but without an explanation as to why this is the case. It is suggested that the possible causes of this be reported in the body of the text, even if they are assumptions.
Reviewer 3 Report
Comments and Suggestions for Authors
The paper evaluate the infection of Rhipicephalus microplus by rickettsial agents in karstik regions. The paper is well organised and I have only few comments.
1) Why do the authors use Wilson scores? In terms of percentages a Fisher exact test (or an extended one) could have been used for testing differences when Chi-square was not convenient? there are also classical methods to evaluate confidence intervals of percentages.
2) Number of ticks tested were different according to the regions. Why?
3) How the different sites were chosen? How they were different?
4) Sample sites are sometime with one name or some others with two, in table1. Why?
5) Co-infections are mentioned very shortly in text and it is a pity. The Supplementary Figure 1 and table 4 provide good information. I suggest that the table should be included in text of the article.
